# Oblivious Sampling Algorithms for Private Data Analysis

**Sajin Sasy**\*
University of Waterloo

**Olga Ohrimenko**
Microsoft Research

## Abstract

We study secure and privacy-preserving data analysis based on queries executed on samples from a dataset. Trusted execution environments (TEEs) can be used to protect the content of the data during query computation, while supporting differential-private (DP) queries in TEEs provides record privacy when query output is revealed. Support for sample-based queries is attractive due to *privacy amplification* since not all dataset is used to answer a query but only a small subset. However, extracting data samples with TEEs while proving strong DP guarantees is not trivial as secrecy of sample indices has to be preserved. To this end, we design efficient secure variants of common sampling algorithms. Experimentally we show that accuracy of models trained with shuffling and sampling is the same for differentially private models for MNIST and CIFAR-10, while sampling provides stronger privacy guarantees than shuffling.

## 1   Introduction

Sensitive and proprietary datasets (e.g., health, personal and financial records, laboratory experiments, emails, and other personal digital communication) often come with strong privacy and access control requirements and regulations that are hard to maintain and guarantee end-to-end. The fears of data leakage may block datasets from being used by data scientists and prevent collaboration and information sharing between multiple parties towards a common good (e.g., training a disease detection model across data from multiple hospitals). For example, the authors of [11, 14, 37] show that machine learning models can memorize individual data records, while information not required for the agreed upon learning task may be leaked in collaborative learning [28]. To this end, we are interested in designing the following secure data query framework:

- A single or multiple *data owners* contribute their datasets to the platform while expecting strong security privacy guarantees on the usage of their data;

- The *framework* acts as a gatekeeper of the data and a computing resource of the data scientist: it can compute queries on her behalf while ensuring that data is protected from third parties;

- *Data scientist* queries the data via the framework via a range of queries varying from approximating sample statistics to training complex machine learning models.

The goal of the framework is to allow data scientist to query the data while providing strong privacy guarantees to data owners on their data. The framework aims to protect against two classes of attackers: the owner of the computing infrastructure of the framework and the data scientist.

The data scientist may try to infer more information about the dataset than what is available through a (restricted) class of queries supported by the framework. We consider the following two collusion scenarios. As the framework may be hosted in the cloud or on premise of the data scientist's

organization, the infrastructure is not trusted as one can access the data without using the query interface. The second collusion may occur in a multi-data-owner scenario where the data scientist could combine the answer of a query and data of one of the parties to infer information about other parties' data. Hence, the attacker may have auxiliary information about the data.

In the view of the above requirements and threat model we propose *Private Sampling-based Query Framework*. It relies on secure hardware to protect data content and restrict data access. Additionally, it supports sample-based differentially private queries for efficiency and privacy. However, naive combination of these components does not lead to an end-to-end secure system for the following reason. Differential privacy guarantees for sampling algorithms (including machine learning model training that build on them [3, 26, 45]) are satisfied only if the sample is hidden. Unfortunately as we will see this is not the case with secure hardware due to leakage of memory access patterns. To this end, we design novel algorithms for producing data samples using two common sampling techniques, Sampling without replacement and Poisson, with the guarantee that whoever observes data access patterns cannot identify the indices of the dataset used in the samples. We also argue that if privacy of data during model training is a requirement then sampling should be used instead of the default use of shuffling since it incurs smaller privacy loss in return to similar accuracy as we show experimentally. We now describe components of our *Private Sampling-based Query Framework*.

**Framework security:** In order to protect data content and computation from the framework host, we rely on encryption and trusted execution environments (TEE). TEEs can be enabled using secure hardware capabilities such as Intel SGX [20] which provides a set of CPU instructions that gives access to special memory regions (enclaves) where encrypted data is loaded, decrypted and computed on. Importantly access to this region is restricted and data is always encrypted in memory. One can also verify the code and data that is loaded in TEEs via attestation. Hence, data owners can provide data encrypted under the secret keys that are available only to TEEs running specific code (e.g., differentially private algorithms). Some of the limitations of TEEs include resource sharing with the rest of the system (e.g., caches, memory, network), which may lead to side-channels [10, 19, 33]. Another limitation of existing TEEs is the amount of available enclave memory (e.g., Intel Skylake CPUs restrict the enclave page cache to 128MB). Though one can use system memory, the resulting memory paging does not only produce performance overhead but also introduces more memory side-channels [44].

**Sample-based data analysis:** Data sampling has many applications in data analysis from returning an approximate query result to training a model using mini-batch stochastic gradient descent (SGD). Sampling can be used for approximating results when performing the computation on the whole dataset is expensive (e.g., graph analysis or frequent itemsets [35, 36]) [2] or not needed (e.g., audit of a financial institution by a regulator based on a sample of the records). We consider various uses of sampling, including queries that require a single sample, multiple samples such as bootstrapping statistics, or large number of samples such as training of a neural network.

Sampling-based queries provide: *Efficiency:* computing on a sample is faster than on the whole dataset, which fits the TEE setting, and can be extended to process dataset samples in parallel with multiple TEEs. *Expressiveness:* a large class of queries can be answered approximately using samples, furthermore sampling (or mini-batching) is at the core of training modern machine learning models. *Privacy:* a query result from a sample reveals information only about the sample and not the whole dataset. Though intuitively privacy may come with sampling, it is not always true. If a data scientist knows indices of the records in the sample used for a query, then given the query result they learn more about records in that sample than about other records. However if sample indices are hidden then there is plausible deniability. Luckily, differential privacy takes advantage of privacy from sampling and formally captures it with *privacy amplification* [8, 21, 25].

**Differential privacy:** Differential privacy (DP) is a rigorous definition of individual privacy when a result of a query on the dataset is revealed. Informally, it states that a single record does not significantly change the result of the query. Strong privacy can be guaranteed in return for a drop in accuracy for simple statistical queries [13] and complex machine learning models [3, 7, 26, 43, 45]. DP mechanisms come with a parameter $\epsilon$, where higher $\epsilon$ signifies a higher privacy loss.

Amplification by sampling is a well known result in differential privacy. Informally, it says that when an $\epsilon$-DP mechanism is applied on a sample of size $\gamma n$ from a dataset $\mathcal{D}$ of size $n$, $\gamma < 1$, then the overall mechanism is $O(\gamma\epsilon)$-DP w.r.t. $\mathcal{D}$. Small $\epsilon$ parameters reported from training of neural networks using DP SGD [3, 26, 45] make extensive use of privacy amplification in their analysis. Importantly, for this to hold they all require the sample identity to be hidden.

DP algorithms mentioned above are set in the trusted curator model where hiding the sample is not a problem as algorithm execution is not visible to an attacker (i.e., the data scientist who obtains the result in our setting). TEEs can be used only as an approximation of this model due to the limitations listed above: revealing memory access patterns of a differentially-private algorithm can be enough to violate or weaken its privacy guarantees. Sampling-based DP algorithms fall in the second category as they make an explicit assumption that the identity of the sample is hidden [42, 24]. If not, amplification based results cannot be applied. If one desires the same level of privacy, higher level of noise will need to be added which would in turn reduce the utility of the results.

Differential privacy is attractive since it can keep track of the privacy loss over multiple queries. Hence, reducing privacy loss of individual queries and supporting more queries as a result, is an important requirement. Sacrificing on privacy amplification by revealing sample identity is wasteful.

**Data-oblivious sampling algorithms** Query computation can be supported in a TEE since samples are small compared to the dataset and can fit into private memory of a TEE. However, naive implementation of data sampling algorithms is inefficient (due to random access to memory outside of TEE) and insecure in our threat model (since sample indices are trivially revealed). Naively hiding sample identity would be to read a whole dataset and only keep elements whose indices happen to be in the sample. This would require reading the entire dataset for each sample (training of models usually requires small samples, e.g., 0.01% of the dataset). This will also not be competitive in performance with shuffling-based approaches used today.

To this end, we propose novel algorithms for producing data samples for two popular sampling approaches: sampling without replacement and Poisson. Samples produced by shuffling-based sampling contain distinct elements, however elements may repeat between the samples. Our algorithms are called data-oblivious [15] since the memory accesses they produce are independent of the sampled indices. Our algorithms are efficient as they require only two data oblivious shuffles and one scan to produce $n/m$ samples of size $m$ that is sufficient for one epoch of training. An oblivious sampling algorithm would be used as follows: $n/m$ samples are generated at once, stored individually encrypted, and then loaded in a TEE on a per-query request.

**Contributions: (i)** We propose a Private Sampling-based Query Framework for querying sensitive data; **(ii)** We use differential privacy to show that sampling algorithms are an important building block in privacy-preserving frameworks; **(iii)** We develop efficient and secure (data-oblivious) algorithms for two common sampling techniques; **(iv)** We empirically show that for MNIST and CIFAR-10 using sampling algorithms for generating mini-batches during differentially-private training achieves the same accuracy as shuffling, even though sampling incurs smaller privacy loss than shuffling.

## 2 Notation and Background

A dataset $\mathcal{D}$ contains $n$ elements; each element $e$ has a key and a value; keys are distinct in $[1, n]$. If a dataset does not have keys, we use its element index in the array representation of $\mathcal{D}$ as a key.

**Trusted Execution Environment** TEE provides strong protection guarantees to data in its private memory: it is not visible to an adversary who can control everything outside of the CPU, e.g., even if it controls the operating system (OS) or the VM. The private memory of TEEs (depending on the side-channel threat model) is restricted to CPU registers (few kilobytes) or caches (32MB) or enclave page cache (128MB). Since these sizes will be significantly smaller than usual datasets, an algorithm is required to store the data in the external memory. Since external memory is controlled by an adversary (e.g., an OS), it can observe its *content* and the *memory addresses* requested from a TEE. Probabilistic encryption can be used to protect the *content* of data in external memory: an adversary seeing two ciphertexts cannot tell if they are encryptions of the same element or a dummy of the same size as a real element.

Though the size of primary memory is not sufficient to process a dataset, it can be leveraged for sample-based data analysis queries as follows. When a query requires a sample, it loads an encrypted

sample from the external memory into the TEE, decrypts it, performs a computation (for example, SGD), discards the sample, and either updates a local state (for example, parameters of the ML model maintained in a TEE) and proceeds to the next sample, or encrypts the result of the computation under data scientist's secret key and returns it.

*Addresses (or memory access sequence)* requested by a TEE can leak information about data. Leaked information depends on adversary's background knowledge (attacks based on memory accesses have been shown for image and text processing [44]). In general, many (non-differentially-private and differentially-private [4]) algorithms leak their access pattern including sampling (see §4.1).

**Data-oblivious algorithms** access memory in a manner that appears to be independent of the sensitive data. For example, sorting networks are data-oblivious as compare-and-swap operators access the same array indices independent of the array content, in contrast to quick sort. Data-oblivious algorithms have been designed for array access [15, 16, 39], sorting [18], machine learning algorithms [32] and several data structures [41]; while this work is the first to consider sampling algorithms. The performance goal of oblivious algorithms is to reduce the number of additional accesses to external memory needed to hide real accesses.

Our sampling algorithms in §4 rely on an oblivious shuffle oblshuffle($\mathcal{D}$) [31]. A shuffle rearranges elements according to permutation $\pi$ s.t. element at index $i$ is placed at location $\pi[i]$ after the shuffle. An oblivious shuffle does the same except the adversary observing its memory accesses does not learn $\pi$. The Melbourne shuffle [31] makes $O(cn)$ accesses to external memory with private memory of size $O(\sqrt[c]{n})$. This overhead is constant since non-oblivious shuffle need to make $n$ accesses. Oblivious shuffle can use smaller private memory at the expense of more accesses (see [34]). It is important to note that while loading data into private memory, the algorithm re-encrypts the elements to avoid trivial comparison of elements before and after the shuffle.

**Differential privacy** A randomized mechanism $\mathcal{M} : D \to \mathcal{R}$ is $(\epsilon, \delta)$ differentially private [13] if for any two neighbouring datasets $\mathcal{D}_0, \mathcal{D}_1 \in D$ and for any subset of outputs $R \in \mathcal{R}$ it holds that $\Pr[\mathcal{M}(\mathcal{D}_0) \in \mathcal{R}] \le e^\epsilon \Pr[\mathcal{M}(\mathcal{D}_1) \in \mathcal{R}] + \delta$. We use substitute-one neighbouring relationship where $|\mathcal{D}_0| = |\mathcal{D}_1|$ and $\mathcal{D}_0, \mathcal{D}_1$ are different in one element. This relationship is natural for sampling without replacement and data-oblivious setting where an adversary knows $|\mathcal{D}|$. As we see in §4.2 hiding the size of Poisson sampling in our setting is non-trivial and we choose to hide the number of samples instead.

Gaussian mechanism [13] is a common way of obtaining differentially private variant of real valued function $f : D \to \mathcal{R}$. Let $\Delta_f$ be the $L_2$-sensitivity of $f$, that is the maximum distance $\|f(\mathcal{D}_0) - f(\mathcal{D}_1)\|_2$ between any $\mathcal{D}_0$ and $\mathcal{D}_1$. Then, Gaussian noise mechanism is defined by $\mathcal{M}(\mathcal{D}) = f(\mathcal{D}) + \mathcal{N}(0, \sigma^2)$ where $\mathcal{N}(0, \sigma^2 \Delta_f^2)$ is a Gaussian distribution with mean 0 and standard deviation $\sigma \Delta_f$. The resulting mechanism is $(\epsilon, \delta)$-DP if $\sigma = \sqrt{2 \log(1.25/\delta)}/\epsilon$ for $\epsilon, \delta \in (0, 1)$.

**Sampling methods** Algorithms that operate on data samples often require more than one sample. For example, machine learning model training proceeds in epochs where each epoch processes multiple batches (or samples) of data. The number of samples $k$ and sample size $m$ are usually chosen such that $n \approx km$ so that every data element has a non-zero probability of being processed during an epoch. To this end, we define $\mathsf{samples}_\mathcal{A}(\mathcal{D}, q, k)$ that produces samples $\mathsf{s}_1, \mathsf{s}_2, \ldots, \mathsf{s}_k$ using a sampling algorithm $\mathcal{A}$ and parameter $q$, where $\mathsf{s}_i$ is a set of keys from $[1, n]$. For simplicity we assume that $m$ divides $n$ and $k = n/m$. We omit stating the randomness used in $\mathsf{samples}_\mathcal{A}$ but assume that every call uses a new seed. We will now describe three sampling methods that vary based on element distribution within each sample and between the samples.

*Sampling without replacement (*SWO*)* produces a sample by drawing $m$ distinct elements uniformly at random from a set $[1, n]$, hence probability of a sample $\mathsf{s}$ is $\frac{1}{n}\frac{1}{n-1} \cdots \frac{1}{n-m+1}$. Let $\mathcal{F}_{\mathsf{SWO}}^{n,m}$ be the set of all SWO samples of size $m$ from domain $[1, n]$; $\mathsf{samples}_{\mathsf{SWO}}(\mathcal{D}, m, k)$ draws $k$ samples from $\mathcal{F}_{\mathsf{SWO}}^{n,m}$ with replacement: elements cannot repeat within the same sample but can repeat between the samples. *Poisson Sampling (*Poisson*)* $\mathsf{s}$ is constructed by independently adding each element from $[1, n]$ with probability $\gamma$, that is $\Pr(j \in \mathsf{s}) = \gamma$. Hence, probability of a sample $\mathsf{s}$ is $\Pr_\gamma(\mathsf{s}) = \gamma^{|\mathsf{s}|}(1 - \gamma)^{n-|\mathsf{s}|}$. Let $\mathcal{F}_{\mathsf{Poisson}}^{n,\gamma}$ be the set of all Poisson samples from domain $[1, n]$. Then, $\mathsf{samples}_{\mathsf{Poisson}}(\mathcal{D}, \gamma, k)$ draws $k$ elements with replacement from $\mathcal{F}_{\mathsf{Poisson}}^{n,\gamma}$. The size of a Poisson sample is a random variable and $\gamma n$ on average. *Sampling via* Shuffle is common for obtaining mini-batches for SGD in practice. It shuffles $\mathcal{D}$ and splits it in batches of size $m$. If more than $k$ samples are required, the procedure is

Table 1: Parameters $(\epsilon', \delta')$ of mechanisms that use $(\epsilon, \delta)$-DP mechanism $\mathcal{M}$ with one of the three sampling techniques with a sample of size $m$ from a dataset of size $n$ and $\gamma = m/n$ for Poisson sampling, where $\epsilon' < 1, \delta'' > 0$, $T$ is the number of samples in an epoch, $E$ is the number of epochs.

| Sampling mechanism | # analyzed samples of size $m$ | |
| --- | --- | --- |
| | $T \leq n/m$ | $T = En/m, E \geq 1$ |
| Shuffling | $\epsilon, \delta$ | $O(\epsilon\sqrt{E\log(1/\delta'')}), E\delta + \delta''$ |
| Poisson, SWO | $O(\epsilon\gamma\sqrt{T\log(1/\delta'')}), T\gamma\delta + \delta''$ | |
| Poisson & Gaussian distribution [3] | $O(\gamma\epsilon\sqrt{T}), \delta$ | |

repeated. Similar to SWO or Poisson, each sample contains distinct elements, however in contrast to them, a sequence of $k$ samples contain distinct elements between the samples.

## 3 Privacy via Sampling and Differential privacy

Privacy amplification of differential privacy captures the relationship of performing analysis over a sample vs. whole dataset. Let $\mathcal{M}$ be a randomized mechanism that is $(\epsilon, \delta)$-DP and let sample be a random sample from dataset $\mathcal{D}$ of size $\gamma n$, where $\gamma < 1$ is a sampling parameter. Let $\mathcal{M}' = \mathcal{M} \circ$ sample be a mechanism that applies $\mathcal{M}$ on a sample of $\mathcal{D}$. Then, informally, $\mathcal{M}'$ is $(O(\gamma\epsilon), \gamma\delta)$-DP [8, 25].

**Sampling** For Poisson and sampling without replacement $\epsilon'$ of $\mathcal{M}'$ is $\log(1 + \gamma(e^\epsilon - 1))$ [25] and $\log(1 + m/n(e^\epsilon - 1))$ [6], respectively. We refer the reader to Balle *et al.* [6] who provide a unified framework for studying amplification of these sampling mechanisms. Crucially all amplification results assume that *the sample is hidden during the analysis as otherwise amplification results cannot hold*. That is, if the keys of the elements of a sample are revealed, $\mathcal{M}'$ has the same $(\epsilon, \delta)$ as $\mathcal{M}$.

Privacy loss of executing a sequence of DP mechanisms can be analyzed using several approaches. Strong composition theorem [13] states that running $T$ $(\epsilon, \delta)$-mechanisms would be $(\epsilon\sqrt{2T\log(1/\delta'')} + T\epsilon(e^\epsilon - 1), T\delta + \delta'')$-DP, $\delta'' \geq 0$. Better bounds can be obtained if one takes advantage of the underlying DP mechanism. Abadi *et al.* [3] introduce a moment account that leverages the fact that $\mathcal{M}'$ uses Poisson sampling and applies Gaussian noise to the output. They obtain $\epsilon' = O(\gamma\epsilon\sqrt{T}), \delta' = \delta$.

**Shuffling** Analysis of differential private parameters of $\mathcal{M}'$ that operates on samples obtained from shuffling is different. Parallel composition by McSherry [27] can be seen as the privacy "amplification" result for shuffling. It states that running $T$ algorithms in parallel on *disjoint* samples of the dataset has $\epsilon' = \max_{i \in [1, T]} \epsilon_i$ where $\epsilon_i$ is the parameter of the $i$th mechanism. It is a significantly better result than what one would expect from using DP composition theorem, since it relies on the fact that samples are disjoint. If one requires multiple passes over a dataset (as is the case with multi-epoch training), strong composition theorem can be used with parallel composition.

**Sampling vs. Shuffling DP Guarantees** We bring the above results together in Table 1 to compare the parameters of several sampling approaches. As we can see sampling based approaches for general DP mechanisms give an order of $O(\sqrt{m/n})$ smaller epsilon than shuffling based approaches. It is important to note that sampling-based approaches assume that the indices (or keys) of the dataset elements used by the mechanism remain secret. In §4 we develop algorithms with this property.

**Differentially private SGD** We now turn our attention to a differentially private mechanism for mini-batch stochastic gradient descent computation. The mechanism is called NoisySGD [7, 38] and when applied instead of non-private mini-batch SGD allows for a release of a machine learning model with differential privacy guarantees on the training data. For example, it has been applied in Bayesian learning [43] and to train deep learning [3, 26, 45] and logistic regression [38] models.

It proceeds as follows. Given a mini-batch (or sample) the gradient of every element in a batch is computed and the L2 norm of the gradient is clipped according to a clipping parameter $C$. Then a noise is added to the sum of the (clipped) gradients of all the elements and the result is averaged over the sample size. The noise added to the result is from Gaussian distribution parametrized with $C$ and a noise scale parameter $\sigma$: $\mathcal{N}(0, \sigma^2 C^2)$. The noise is proportional to the sensitivity of the sum of gradients to the value of each element in the sample. The amount of privacy budget that a single

batch processing, also called subsampled Gaussian mechanism, incurs depends on the parameters of the noise distribution and how the batch is sampled. The model parameters are iteratively updated after every NoisySGD processing. The number of iterations and the composition mechanism used to keep track of the privacy loss determine the DP parameters of the overall training process.

Abadi *et al.* [3] report analytical results assuming Poisson sampling but use shuffling to obtain the samples in the evaluation. Yu *et al.* [45] point out the discrepancy between analysis and experimental results in [3], that is the reported privacy loss is underestimated due to the use of shuffling. Yu *et al.* proceed to analyze shuffling and sampling but also use shuffling in their experiments. Hence, though analytically Poisson and SWO sampling provide better privacy parameters than shuffling, there is no evidence that the accuracy is the same between the approaches in practice. We fill in this gap in §5 and show that for the benchmarks we have tried it is indeed the case.

# 4 Oblivious Sampling Algorithms

In this section, we develop data-oblivious algorithms for generating a sequence of samples from a dataset $\mathcal{D}$ such that the total number of samples is sufficient for a single epoch of a training algorithm. Moreover, our algorithms will access the original dataset at indices that appear to be independent of how elements are distributed across the samples. As a result, anyone observing their memory accesses cannot identify, how many and which samples each element of $\mathcal{D}$ appears in.

## 4.1 Oblivious sampling without replacement (SWO)

We introduce a definition of an *oblivious sampling algorithm*: oblivious samples$_{\mathsf{SWO}}(\mathcal{D}, m)$ is a randomized algorithm that returns $k$ SWO samples from $\mathcal{D}$ and produces memory accesses that are indistinguishable between invocations for all datasets of size $n = |\mathcal{D}|$ and generated samples.

As a warm-up, consider the following naive way of generating a single SWO sample of size $m$ from dataset $\mathcal{D}$ stored in external memory of a TEE: generate $m$ distinct random keys from $[1, n]$ and load from external memory elements of $\mathcal{D}$ that are stored at those indices. This trivially reveals the sample to an observer of memory accesses. A secure but inefficient way would be to load $\mathcal{D}[l]$ for $\forall l \in [1, n]$ and, if $l$ matches one of the $m$ random keys, keep $\mathcal{D}[l]$ in private memory. This incurs $n$ accesses to generate a sample of size $m$. Though our algorithm will also make a linear number of accesses to $\mathcal{D}$, it will amortize this cost by producing $n/m$ samples.

The high level description of our secure and efficient algorithm for producing $k$ is as follows. Choose $k$ samples from $\mathcal{F}_{\mathsf{SWO}}^{n,m}$, numbering each sample with an identifier $1$ to $k$; the keys within the samples (up to a mapping) will represent the keys of elements used in the samples of the output. Then, while scanning $\mathcal{D}$, replicate elements depending on how many samples they should appear in and associate each replica with its sample id. Finally, group elements according to sample ids.

**Preliminaries**  Our algorithm relies on a primitive that can efficiently draw $k$ samples from $\mathcal{F}_{\mathsf{SWO}}^{n,m}$ (denoted via SWO.initialize$(n, m)$). It also provides a function SWO.samplemember$(i, j)$ that returns True if key $j$ is in the $i$th sample and False otherwise. This primitive can be instantiated using $k$ pseudo-random permutations $\rho_i$ over $[1, n]$. Then sample $i$ is defined by the first $m$ indices of the permutation, i.e., element with key $j$ is in the sample $i$ if $\rho_i(j) \leq m$. This procedure is described in more detail in Appendix §A.

We will use $r_j$ to denote the number of samples where key $j$ appears in, that is $r_j = |\{i \mid \mathsf{samplemember}(i, j), \forall i \in [1, k], \forall j \in [1, n]\}|$. It is important to note that samples drawn above are used as a template for a valid SWO sampling (i.e., to preserve replication of elements across the samples). However, the final samples $\mathsf{s}_1, \mathsf{s}_2, \ldots, \mathsf{s}_k$ returned by the algorithm will be instantiated with keys that are determined using function $\pi'$ (which will be defined later). In particular, for all samples, if samplemember$(i, j)$ is true then $\pi'(j) \in \mathsf{s}_i$.

**Description**  The pseudo-code in Algorithm 1 provides the details of the method. It starts with dataset $\mathcal{D}$ obliviously shuffled according to a random secret permutation $\pi$ (Line 1). Hence, element $e$ is stored (re-encrypted) in $\mathcal{D}$ at index $\pi(e.\mathsf{key})$. The next phase replicates elements such that for every index $j \in [1, n]$ there is an element (not necessarily with key $j$) that is replicated $r_j$ times (Lines 4-14). The algorithm maintains a counter $l$ which keeps the current index of the scan in the array and $e_{\mathsf{next}}$ which stores the element read from $l$th index.

Additionally the algorithm maintains element $e$ which is an element that currently is being replicated. It is updated to $e_{next}$ as soon as sufficient number of replicas is reached. The number of times $e$ is replicated depends on the number of samples element with key $j$ appears in. Counter $j$ starts at 1 and is incremented after element $e$ is replicated $r_j$ times. At any given time, counter $j$ is an indicator of the number of distinct elements written out so far. Hence, $j$ can reach $n$ only if every element appears in exactly one sample. On the other hand, the smallest $j$ can be is $m$, this happens when all $k$ samples are identical.

Given the above state, the algorithm reads an element into $e_{next}$, loops internally through $i \in [1..k]$: if current key $j$ is in $i$th sample it writes out an encrypted tuple $(e, i)$ and reads the next element from $\mathcal{D}$ into $e_{next}$. Note that $e$ is re-encrypted every time it is written out in order to hide which one of the elements read so far is being written out. After the scan, the tuples are obliviously shuffled. At this point, the sample id $i$ of each tuple is decrypted and used to (non-obliviously) group elements that belong to the same sample together, creating the sample output $s_1..s_k$ (Lines 16-20).

We are left to derive the mapping $\mathsf{m}$ between keys used in samples drawn in Line 2 and elements returned in samples $s_1..s_k$. We note that $\mathsf{m}$ is not explicitly used during the algorithm and is used only in the analysis. From the algorithm we see that $\mathsf{m}(l) = \pi^{-1}(1 + \sum_{j=1}^{l-1} r_j)$, that is $\mathsf{m}$ is derived from $\pi$ with shifts due to replications of preceding keys. (Observe that if every element appears only in one sample $\mathsf{m}(l) = \pi^{-1}(l)$.) We show that $\mathsf{m}$ is injective and random (Lemma 1) and, hence, $s_1..s_k$ are valid SWO samples.

**Algorithm 1** Oblivious samples$_{\mathsf{SWO}}(\mathcal{D}, m)$: takes an encrypted dataset $\mathcal{D}$ and returns $k = n/m$ SWO samples of size $m$, $n = |\mathcal{D}|$.

1: $\mathcal{D} \leftarrow$ oblshuffle$(\mathcal{D})$
2: SWO.initialize$(n, m)$
3: $\mathcal{S} \leftarrow [], j \leftarrow 1, l \leftarrow 1, e \leftarrow \mathcal{D}[1],$
    $e_{next} \leftarrow \mathcal{D}[1]$
4: **while** $l \leq n$ **do**
5:   **for** $i \in [1, k]$ **do**
6:     **if** SWO.samplemember$(i, j)$ **then**
7:       $\mathcal{S}$.append(re-enc$(e)$, enc$(i)$)
8:       $l \leftarrow l + 1$
9:       $e_{next} \leftarrow \mathcal{D}[l]$
10:     **end if**
11:   **end for**
12:   $e \leftarrow e_{next}$
13:   $j \leftarrow j + 1$
14: **end while**
15: $S \leftarrow$ oblshuffle$(\mathcal{S})$
16: $\forall i \in [1, k] : s_i \leftarrow []$
17: **for** $p \in S$ **do**
18:   $(c_e, c_i) \leftarrow p, i \leftarrow$ dec$(c_i)$
19:   $s_i \leftarrow s_i$.append$(c_e)$
20: **end for**
21: Return $s_1, s_2, \ldots, s_k$

**Example** Let $\mathcal{D} = \{(1, A), (2, B), (3, C), (4, D), (5, E), (6, F)\}$, where $(4, D)$ denotes element $D$ at index 4 (used also as a key), $m = 2$, and randomly drawn samples in SWO.initialize are $\{1, 4\}$, $\{1, 2\}$, $\{1, 5\}$. Suppose $\mathcal{D}$ after the shuffle is $\{(4, D), (1, A), (5, E), (3, C), (6, F), (2, B)\}$. Then, after the replication $\mathcal{S} = \{((4, D), 1), ((4, D), 2), ((4, D), 3), ((3, C), 2), ((6, F), 1), ((2, B), 3)\}$ where the first tuple $((4, D), 1)$ indicates that $(4, D)$ appears in the first sample.

**Correctness** We show that samples returned by the algorithm correspond to samples drawn randomly from $\mathcal{F}_{\mathsf{SWO}}^{m,n}$. We argue that samples returned by the oblivious samples$_{\mathsf{SWO}}$ are identical to those drawn truly at random from $\mathcal{F}_{\mathsf{SWO}}^{m,n}$ up to key mapping $\mathsf{m}$ and then show that $\mathsf{m}$ is injective and random in Appendix A. For every key $j$ present in the drawn samples there is an element with key $\mathsf{m}(j)$ that is replicated $r_j$ times and is associated with the sample ids of $j$. Hence, returned samples, after being grouped, are exactly the drawn samples where every key $j$ is substituted with an element with key $\mathsf{m}(j)$.

**Security and performance** The adversary observes an oblivious shuffle, a scan where an element is read and an encrypted pair is written, another oblivious shuffle and then a scan that reveals the sample identifiers. All patterns except for revealing of the sample identifiers are independent of the data and sampled keys. We argue security further in §A. Performance of oblivious SWO sampling is dominated by two oblivious shuffles and the non-oblivious grouping, replication scan has linear cost. Hence, our algorithm produces $k$ samples in time $O(cn)$ with private memory of size $O(\sqrt[c]{n})$. Since a non-oblivious version would require $n$ accesses, our algorithm has a constant overhead for small $c$.

**Observations** We note that if more than $k$ samples of size $m = n/k$ need to be produced, one can invoke the algorithm multiple times using different randomness. Furthermore, Algorithm 1 can produce samples of varying sizes $m_1, m_2, .., m_k$ ($n = \sum m_i$) given as an input. The algorithm itself will remain the same. However, in order to determine if $j$ is in sample $i$ or not, samplemember$(i, j)$ will check if $\rho_i(j) \leq m_i$ instead of $\rho_i(j) \leq m$.

## 4.2 Oblivious Poisson sampling

Performing Poisson sampling obliviously requires not only hiding access pattern but also the size of the samples. Since in the worst case the sample can be of size $n$, each sample will need to be padded to $n$ with dummy elements. Unfortunately generating $k$ samples each padded to size $n$ is impractical. Though samples of size $n$ are unlikely, revealing some upper bound on sample size would affect the security of the algorithms relying on Poisson sampling.

Instead of padding to the worst case, we choose to hide the number of samples that are contained within an $n$-sized block of data (e.g., an epoch). In particular, our oblivious Poisson sampling returns $S$ that consists of samples $\mathsf{s}_1, \mathsf{s}_2, \ldots, \mathsf{s}_{k'}$ where $k' \leq k$ such that $\sum_{i \in [1,k']} |\mathsf{s}_i| \leq n$. The security of sampling relies on hiding $k'$ and the boundary between the samples, as otherwise an adversary can estimate sample sizes.

The algorithm (presented in Appendix §B) proceeds similar to SWO except every element, in addition to being associated with a sample id, also stores its position in final $S$. The element and the sample id are kept private while the position is used to order the elements. It is then up to the queries that operate on the samples inside of a TEE (e.g., SGD computation) to use sample id while scanning $S$ to determine the sample boundaries. The use of $\mathsf{samples}_{\mathsf{Poisson}}$ by the queries has to be done carefully without revealing when the sample is actually used as this would reveal the boundary (e.g., while reading the elements during an epoch, one needs to hide after which element the model is updated).

We assume that that samples from $\mathcal{F}_{\mathsf{Poisson}}^{n,\gamma}$ can be drawn efficiently and describe how in Appendix §B. The algorithm relies on two functions that have access to the samples: $\mathsf{getsamplesize}(i)$ and $\mathsf{getsamplepos}(i,l)$ which return the size of the $i$th sample and the position of element $l$ in $i$th sample. The algorithm uses the former to compute $k'$ and creates replicas for samples with identifiers from 1 to $k'$. The other changes to the Algorithm 1 are that $\mathcal{S}.\mathsf{append}(\mathsf{enc}(e), \mathsf{enc}(i))$ is substituted with $\mathcal{S}.\mathsf{append}(\mathsf{enc}(e), \mathsf{enc}(i), \mathsf{enc}(\mathsf{pos}))$ where $\mathsf{pos} = \sum_{i' < i} \mathsf{getsamplesize}(i') + \mathsf{getsamplepos}(i,l)$. If the total number of elements in the first $k'$ samples is less than $n$, the algorithm appends dummy elements to $\mathcal{S}$. $\mathcal{S}$ is then shuffled. After that positions pos can be decrypted and sorted (non-obliviously) to bring elements from the same samples together. In a decrypted form this corresponds to samples ordered one after another sequentially, following with dummy elements if applicable.

## 5 Experimental results

The goal of our evaluation is to understand the impact of sampling on the accuracy of training of neural network models and their differentially private variants, We show that accuracy of all sampling mechanisms is the same while shuffling has the highest privacy loss.

We use TensorFlow v1.13 and TensorFlow Privacy library [5] for DP training. We implement non-oblivious SWO and Poisson sampling mechanisms since accuracy of the training procedure is independent of sampling implementation. We report an average of 5 runs for each experiment.

Our implementation relies on DP optimizer from [5] which builds on ideas from [3] to implement noisySGD as described in §3. Note that this procedure is independent of the sampling mechanism behind how the batch is obtained. The only exception is Poisson where the average is computed using a fixed sample size ($\gamma \times n$) vs. its real size as for the other two sampling mechanisms. We set the clipping parameter to 4, $\sigma = 6$, $\delta = 10^{-5}$. For each sampling mechanism we use a different privacy accountant to compute exact total $\epsilon$ as opposed to asymptotical guarantees in Table 1. For shuffling we use [45, 27]; for Poisson sampling [5]; and for SWO we implement the approach from [42].

**MNIST** dataset contains 60,000 train and 20,000 test images of ten digits with the classification tasks of determining which digit an image corresponds to. We use the same model architecture as [3] and [45]. It is a feed-forward neural network comprising of a single hidden layer with 1000 ReLU units and the output layer is softmax of 10 classes corresponding to the 10 digits. The loss function computes cross-entropy loss. During training we sample data using shuffling, sampling without replacement and Poisson. For the first two we use batch size $m = 600$, $\gamma = 0.01$ and $m = 200$, $\gamma = 0.003$ in Figure 1. Each network is trained for 100 epochs. We report the results in Table 2 (left). We observe that sampling mechanism does not change accuracy for this benchmark.

**CIFAR-10** dataset consists of 50,000 training and 10,000 test color images classified into 10 classes [1]. Each example is a $32 \times 32$ image with three channels (RGB). We use the training setup from the TensorFlow tutorial [2] for CIFAR-10 including the data augmentation step. The

Table 2: Test (Train) accuracy of MNIST & CIFAR10 models trained with samples generated with Shuffle, Poisson and sampling w/o replacement (SWO) and their differentially private (DP) variants with incurred total $\epsilon$.

|  | Shuffle | Poisson | SWO |  | Shuffle | SWO |
|---|---|---|---|---|---|---|
| MNIST | 97.5 (98.33) | 97.47 (98.31) | 97.43 (98.31) | CIFAR-10 | 79.6 (83.2) | 79 (82.9) |
| DP MNIST | 94.06 (94.1) | 94.1 (94.01) | 94.03 (94.05) | DP CIFAR-10 | 73.4 (72.3) | 72.5 (71) |
| $\epsilon$ | 9.39 | 0.82 | 2.13 | $\epsilon$ | 9.39 | 4.89 |

same setup was also used in [3]. The network consists of two convolutional layers followed by two fully connected layers. Similar to [3, 45] we use a public dataset (CIFAR-100) to train a network with the same architecture. We then use the pre-trained network to train the fully connected layers using the CIFAR-10 dataset. Each network is trained for 100 epochs with sample size of $m = 2000$.

We use the same network setup as related work [3]; but better accuracy can be achieved with deeper networks. The results for shuffling and sampling w/o replacement are in Table 2 (right). Similar to MNIST there is no significant difference between the two.

**Sampling in differentially private training** In Table 2 (middle row) we compare the effect of sampling approaches on DP training. Similar to results reported in previous work DP training degrades model performance. However, accuracy between sampling approaches is similar. The difference between the sampling mechanism is evident however in the total privacy loss they occur. The results in last row of Table 2 show that shuffling incurs the highest privacy loss for the same number of epochs, in line with asymptotical guarantees in Table 1. In Figure 1 we show that as expected smaller sample (batch) size has a positive effect on $\epsilon$ for sampling.

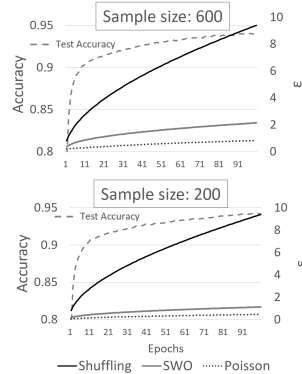

Figure 1: Accuracy and $\epsilon$ for MNIST over epochs for sample sizes 200 and 600.

These results indicate that if maintaining low privacy loss is important then SWO and Poisson should be the preferred option for obtaining batches: sampling gives smaller privacy loss and same accuracy.

## 6 Related work

The use of TEEs for privacy-preserving data analysis has been considered in several prior works. Multi-party machine learning using Intel SGX and data-oblivious machine learning algorithms has been described in [32]. PROCHLO [9] shuffles user records using TEEs for anonymization. Secret shuffle allows PROCHLO to obtain strong guarantees from local DP algorithms [24] that are applied to records before the shuffle. Systems in [46, 30] consider map-reduce-like computation for data analysis while hiding access pattern between computations. Slalom [40] proposes a way to partially outsource inference to GPUs from TEEs while maintaining integrity and privacy.

Oblivious algorithms as software protection were first proposed in [15, 16]. Recently, relaxation of security guarantees for hiding memory accesses have been considered in the context of differential privacy. Allen *et al.* [4] propose an oblivious differentially-private framework for designing DP algorithms that operate over data that does not fit into private memory of a TEE (as opposed to sample-based analysis). Chan *et al.* [12] have considered implications of relaxing the security guarantees of hiding memory accesses from data-oblivious definition to the differentially-private variant. Neither of these works looked at the problem of sampling.

We refer the reader to [13] for more information on differential privacy. Besides work mentioned in §3, we highlight several other works on the use of sampling for differential privacy. Sample-Aggregate [29] is a framework based on sampling where $k$ random samples are taken such that in total all samples have $\approx n$ elements, a function is evaluated on each sample, and $k$ outputs are then aggregated and reported with noise. Kasiviswanathan *et al.* [22] study concept classes that can be learnt in differentially private manner based on a sample size and number of interactions. DP natural language models in [26] are trained using a method of [3] while using data of a single user as a mini-batch. Amplification by sampling has been studied for Rényi differential private mechanisms in [42]. Finally, PINQ [27], assuming a trusted curator setting, describes a system for answering database queries with DP guarantees.

## Acknowledgements

The authors would like to thank anonymous reviewers for useful feedback that helped improve the paper. The authors are also grateful to Marc Brockschmidt, Jana Kulkarni, Sebastian Tschiatschek and Santiago Zanella-Béguelin for insightful discussions on the topic of this work.

## Footnotes

\*Work done during internship at Microsoft Research.

[2] We note that we use sampling differently from statistical approaches that treat the dataset $\mathcal{D}$ as a sample from a population and use all records in $\mathcal{D}$ to estimate parameters of the underlying population.

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
