[Supplementary Material · camera-ready-full.pdf]

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

# A Details of Oblivious SWO Sampling (§4.1)

**Sampling primitive** A single sample of SWO from domain $[1, n]$ can be instantiated using a permutation $\rho$ over $[1, n]$. The sample is defined by elements that are mapped to the first $m$ elements, i.e., element $j$ is in the sample if $\rho(j) \leq m$. This procedure is described in Algorithm 2 for $k$ samples. During the initialize call, random permutations are chosen (e.g., in the real implementation this would correspond to choosing a random seed and then deriving $k$ seeds for each permutation). Then, samplemember$(i, j)$ returns True or False depending on whether $j$ is in the $i$th sample or not.

Observe that each sample defined by the above primitive represents a valid SWO sample. Let $\mathsf{s}_i$ be the sample that consists of the first $m$ elements of the permutation $\rho_i$. The probability of choosing a particular sample is the probability of choosing one of the permutations where the first $m$ elements are fixed. Since there are $(n - m)!$ permutations with first $m$ elements fixed: the probability of $\mathsf{s}_i$ is $(n - m)!/n!$ and is $\frac{1}{n}\frac{1}{n-1}\cdots\frac{1}{n-m+1}$ which is the probability of an SWO sample.

---

**Algorithm 2** Instantiation of SWO sampling for $k = n/m$ samples drawn from $\mathcal{F}_{\mathsf{SWO}}^{n,m}$

---

    initialize$(n, m)$: choose random permutations with domain $[1, n]$: $\rho_1, \rho_2, \ldots, \rho_k$
    samplemember$(i, j)$: If $\rho_i(j) \leq m$ return True, else False

---

**Security of Algorithm 1** The adversary observes an oblivious shuffle, a scan where an element is read and an encrypted pair is written, another oblivious shuffle and then a scan that reveals the sample identifiers. Since oblivious shuffle is independent of the content of $\mathcal{D}$ and the shuffle permutation, all patterns except for revealing of the sample identifiers are independent of the data. We are left to argue that revealing sample ids and their locations (i.e., indices in the output $S$) does not reveal information about the data nor the samples. First note that there are $m$ copies of sample ids $1, 2, \ldots, k$ associated with a ciphertext, hence data-independent. Second, note that locations of the revealed identifiers are random according to the permutation chosen in the second shuffle step. Since the permutation of the shuffles are hidden, the adversary does not learn the location of the tuple before and after the shuffle.

**Lemma 1.** *Let $\pi$ be a permutation over $n$ elements, $\forall j \in [1, n], r_j \in [0, k]$ such that $\sum^n r_j = n$ and $\mathcal{K} = \{j \mid r_j \geq 1\}$. For $l \in [1, n]$, let $\mathsf{m}(l) = \pi^{-1}(1 + \sum_{j=1}^{l-1} r_j)$. Then $\mathsf{m}$ evaluated on keys in $\mathcal{K}$ is an injective random function over $[1, n]$.*

*Proof.* The statement follows from two observations: $\pi^{-1}$ is a permutation and $\pi^{-1}$ is evaluated only on distinct elements from a set $[1, n]$.

The second observation is true since the mapping from $l$ to $1 + \sum_{j=1}^{l-1} r_j$, when evaluated on $l \in [1, n]$, is injective as it is strictly monotonic. Moreover, $(1 + \sum_{j=1}^{l-1} r_j) \leq n$ since $\sum^n r_j = n$.

Co-domain of $\mathsf{m}$ appears independent of its input since it is a subset of the output of a random permutation function $\pi$ that has these properties by definition. □

# B Details of Oblivious Poisson Sampling (§4.2)

**Sampling primitive** Instantiation of Poisson samples (Algorithm 3) is an extension of SWO sampling that in addition also randomly chooses the size for each sample, $M_i$. Recall that for SWO the sample size is fixed ($m$) while Poisson sampling takes $\gamma$ as a parameter and adds an element to the sample with probability $\gamma$. Since the size of a Poisson sample is a random variable $\mathsf{Binom}(n, \gamma)$, for each sample we draw a random variable from this Binomial distribution and use it to determine the sample size.

Observe that each sample defined by the above primitive represents a valid Poisson sample. Let $\mathsf{s}_i$ be the sample that consists of the first $M_i$ elements of the permutation $\rho_i$. The probability of choosing $\mathsf{s}_i$ is the probability of the Binomial random variable being $M_i$ and then choosing a permutation where the first $M_i$ elements are fixed. The probability of the former is $\binom{n}{M_i}\gamma^{M_i}(1 - \gamma)^{M_i}$. Then the probability of $\mathsf{s}_i$ is $\binom{n}{M_i}\gamma^{M_i}(1 - \gamma)^{M_i}(n - M_i)!/n! = \gamma^{M_i}(1 - \gamma)^{M_i}/M_i!$ and is $\gamma^{M_i}(1 - \gamma)^{M_i}$ if the element order within the sample is not relevant. Hence, samples produced by Algorithm 3 are distributed as Poisson samples of the corresponding sizes.

---

**Algorithm 3** Instantiation of Poisson sampling for $k = n\gamma$ samples drawn from $\mathcal{F}_{\mathsf{Poisson}}^{n,\gamma}$

---

    $\mathsf{initialize}(n, \gamma)$:
      choose random permutations with domain $[1, n]$: $\rho_1, \rho_2, \ldots, \rho_k$
      $\forall i \in [1, k]$, $M_i \leftarrow \mathsf{Binom}(n, \gamma)$
    $\mathsf{samplemember}(i, j)$: If $\rho_i(j) \leq M_i$ return True, else False
    $\mathsf{getsamplesize}(i)$: return $M_i$
    $\mathsf{getsamplepos}(i, l)$: return $\rho_i(l)$

---

---

**Algorithm 4** Oblivious $\mathsf{samples}_{\mathsf{Poisson}}(\mathcal{D}, \gamma)$: takes an encrypted dataset $\mathcal{D}$ and returns Poisson sample(s) with parameter $\gamma$, $n = |\mathcal{D}|$

---

  1: $\mathcal{D} \leftarrow \mathsf{oblshuffle}(\mathcal{D})$
  2: $\mathsf{Poisson.initialize}(n, \gamma)$
  3: $\mathcal{S} \leftarrow []$
  4: $j \leftarrow 1$, $l \leftarrow 1$, $e \leftarrow \mathcal{D}[1]$, $e_{\mathsf{next}} \leftarrow \mathcal{D}[1]$
  5: $k' \leftarrow 1$, $\mathsf{cursize} \leftarrow \mathsf{Poisson.getsamplesize}(1)$
  6: **while** $\mathsf{cursize} + \mathsf{Poisson.getsamplesize}(k' + 1) \leq n$ and $k' + 1 \leq k$ **do**
  7:    $k' \leftarrow k' + 1$
  8:    $\mathsf{cursize} \leftarrow \mathsf{cursize} + \mathsf{Poisson.getsamplesize}(k')$
  9: **end while**
10: **while** $j \leq \mathsf{cursize}$ **do**
11:    **for** $i \in [1, k']$ **do**
12:      **if** $\mathsf{Poisson.samplemember}(i, l)$ **then**
13:         $\mathsf{pos} \leftarrow \sum_{i' < i} \mathsf{Poisson.getsamplesize}(i') + \mathsf{Poisson.getsamplepos}(i, l)$
14:         $\mathcal{S}.\mathsf{append}(\mathsf{re\text{-}enc}(e), \mathsf{enc}(i), \mathsf{enc}(\mathsf{pos}))$
15:         $j \leftarrow j + 1$
16:         $e_{\mathsf{next}} \leftarrow \mathcal{D}[j]$
17:      **end if**
18:    **end for**
19:    $e \leftarrow e_{\mathsf{next}}$
20:    $l \leftarrow l + 1$
21: **end while**
22: **for** $j \in [\mathsf{cursize} + 1, n]$ **do**
23:    $\mathcal{S}.\mathsf{append}(\mathsf{enc}(\mathsf{dummy}), \mathsf{enc}(0), \mathsf{enc}(j))$
24: **end for**
25: $S \leftarrow \mathsf{oblshuffle}(\mathcal{S})$
26: Decrypt pos (last part) of every tuple in $S$ and use it to sort the encrypted elements
27: Return $S$

---

**Analysis** Performance of oblivious Poisson sampling is dominated by two oblivious shuffles and the non-oblivious sorting in Line 26 since the replication scan is linear.

Security of Algorithm 4 follows that of SWO except it requires $k'$ to be hidden from an adversary. In particular, the adversary observes two shuffles and a scan where one element is read and one written out (hence, data-independent). The content of the elements written out is encrypted or re-encrypted to hide which elements read from external memory are written out. The total size of all the samples (cursize) is protected by padding the output to $n$. Hence, the adversary observes only positions of every tuple (Line 26). However, by construction position values are 1 to $n$ and are randomly shuffled, hence, they are independent of the actual samples. The sample boundary, which can be determined from the middle part of the tuples, $\mathsf{enc}(i)$, in $\mathcal{S}$, is encrypted and not revealed to the adversary. Note that tuples with $\mathsf{enc}(0)$ denote padded dummy elements and do not belong to any sample.