[Reviews · NeurIPS 2019]

Reviewer 1



+ The proposed solution in the paper is important as it provides a practical way to implement the privacy amplification methods in the literature. + The proposed solution is simple and is reminiscent of the bag of little bootstrap paper https://arxiv.org/pdf/1112.5016.pdf . Q: Where else other than learning problems this framework can be used? And it would be great if the authors do an empirical study of that.

Reviewer 2



As mentioned above, the proposed oblivious sampling schemes are quite straightforward, and basically rely on (existing) oblivious data shuffling techniques along with some careful replication of data elements. The results appear to be novel, but not particularly original or technically deep. In addition, the schemes seem to work only for the case where k=n/m samples of size m each are needed from a data set of size n. Not clear if/how they might be extended to more general sampling scenarios. The paper is overall clearly written, modulo a few typos and bugs spread throughout the text (e.g., delta vs delta_f, eps vs eps' on page 4). Some parts need a more thorough discussion and explanation, e.g., the various symbols and formulas in the comparisons with the shuffling approach in Table 1. The proposed sampling algorithms are empirically shown to provide comparable utility with stronger DP guarantees for noisySGD over two test data sets. Given the limited scope of the experiments, it is not clear that these this can be extrapolated to other data sets and/or learning tasks.

Reviewer 3



Update: I thank the authors for their responses to my fellow reviewers. My opinion remains unchanged and I still recommend acceptance. Significance: The problem investigated is important to the development of secure and private differential privacy in practice Clarity: The writing contains some grammatical mistakes and typos but they are small. Overall it is clear. Originality: They consider the new problem of designing oblivious versions of sampling without replacement and poisson sampling. This seems like a unique and important problem, but I am unfamiliar with the area.

Reviewer 4



The main contributions of the work are to provide 2 algorithms for sampling in a Trusted Execution Environment (TEE). TEEs are helpful for protecting the privacy of the input data during a query computation, whereas sampling is advantageous in protecting the privacy of the input from seeing the output of a differentially private (DP) query. The authors also provide an evaluation of both the sampling techniques on MNIST and CIFAR with DP SGD. Originality: Regarding both the proposed algorithms, they heavily rely on a data oblivious shuffle. Given that the oblivious shuffle is from prior work, the novelty of the proposed algorithms seems incremental at best. Moreover, it is not clear why naively sampling multiple batches after a data oblivious shuffle within a TEE will be any worse than the proposed algorithms (as the shuffle will effectively hide which elements are present in the samples, even though memory accesses might be observable). The empirical evaluation needs to be broader and deeper to have a significant contribution. Another minor point is that the first 2 contributions stated in the contributions paragraph (117-122) are already known in the differentially privacy literature, and the authors have provided citations of many such works. Quality: The submission is technically sound. The claims are supported by theoretical analysis. Clarity: The quality of writing is not satisfactory, as even the only example they describe for explaining their 1st algorithm (lines 287-291) is poorly written, and I suspect it contains some typos. There are also other typos in the paper (e.g., line 185, 243). Significance: The significance of the proposed algorithms seems minor given prior work. ---------------------------------- Edit after author response: Update on Originality: After reading the author response and other reviews, I think that the contribution of the proposed algorithms over prior works is non-trivial. It is also clear to me why naive sampling after an oblivious shuffle will not share the privacy amplification advantage that the proposed algorithms have. Update on Significance: Impactful.

[Author Response · NeurIPS 2019]

We thank the reviewers for their thorough and comprehensive reviews. We have condensed our responses on what we deemed to be the most important highlighted points.

**Reviewer 2** "Where else other than learning problems this framework can be used?"

Our introduction can be expanded with the following known applications that rely on estimates from sampling (in our case, they would be set in a secure environment and provide security and privacy): statistical quality control, statistical data analysis of large datasets (e.g., graphs), auditing (e.g., a financial institution can allow a regulator/data scientist to analyze only several samples from its data).

"More empirical analysis." Related work on NoisySGD [3,45] also used CIFAR10 and MNIST in the experiments. Given the opportunity, we are happy to add our experiments on categorical data.

**Reviewer 3** "The proposed algorithms are simple, and rely on employing oblivious shuffling along with careful replication of elements. From a technical perspective, the techniques and their analysis is rather straightforward. The results appear to be novel, but not particularly original or technically deep."

- We believe that simplicity is not a disadvantage when designing security algorithms since it minimizes a potential for an implementation error and, hence, a vulnerability;
- Simplicity may increase chances of adoption among practitioners. One key contribution of widely used Path ORAM is its simplicity compared to first oblivious array schemes based on "sophisticated deamortized oblivious sorting and oblivious cuckoo hash table" ["Path ORAM: An Extremely Simple Oblivious RAM Protocol" by Stefanov *et al.*];
- Finally, though simple, it was not straightforward to simulate the same element distribution as SWO and Poisson in oblivious manner (i.e., we proposed to rely on pseudo-random permutation to simulate the sampling distribution) *and* to do so in one pass over the data (our design involved substituting real keys with those drawn by the permutation).

"...the schemes seem to work only for the case where k=n/m samples of size m each are needed from a data set of size n. Not clear if/how they might be extended to more general sampling scenarios."

We thank the reviewer for pointing out general sampling cases. In fact, our algorithm can support them:

- Given a sequence of sample sizes $m_1, m_2, .., m_k$ ($n = \sum m_i$), Alg. 1 will not change. However, samplemember$(i, j)$ (Alg. 2, Appendix) will check if $\rho_i(j) \leq m_i$ instead of $\rho_i(j) \leq m$ to determine if $j$ is in sample $i$ or not.
- For $k > n/m$ one can invoke the algorithm again and for $k < n/m$ one can ignore additional samples.
- $k = n/m$ or $n = \sum_k m_i$ achieve the best performance since the cost of shuffling is amortized.

**Reviewer 4** We are grateful to the reviewer for their positive feedback and pointing out the significance and originality of our work. We are happy to integrate related work with the introduction.

**Reviewer 5** "1. Clearly state benefits over naive sampling multiple batches in a TEE after an oblivious shuffle."

The naive sampling would incur $(\epsilon\sqrt{2T \log(1/\delta')} + T\epsilon(e^\epsilon - 1), T\delta + \delta')$-DP after $T$ samples. Both of the DP parameters for our sampling approaches will be $O(\gamma) \times$ smaller, where $\gamma = m/n \leq 1$ and $m$ is the sample size.

DP Analysis: The algorithm suggested by the reviewer cannot rely on the $\gamma$ "privacy amplification" since the adversary is able to observe which elements (up to a permutation) appear in which samples (e.g., how many and which samples an element appears in and what is the co-occurrence of elements across the samples). Even though it does not know which element it is exactly, the access pattern reveals overlaps between the samples; overall this is not sufficient for hiding sample identity. Hence, to analyze it, one has to rely on strong composition theorem, leading to DP guarantees that are worse than algorithms from Table 1. (Shuffling in Table 1 relies on the fact that shuffled samples do not overlap.)

"2. ...the authors can evaluate the performance of the techniques after tuning at the same privacy level (rather than for the same number of epochs). ...how tuning has been done.... It is unclear that the best values for learning rate and minibatch size will be the same for all the evaluated algorithms."

In Figure 1 we show privacy-accuracy tradeoff for sample sizes 200 and 600. For example, for size 200: with privacy level fixed at $\epsilon \leq 0.81$: the accuracy is 94% for Poisson ($\epsilon = 0.47$), 90% for SWO ($\epsilon = 0.47$), and 78% for Shuffle ($\epsilon = 0.81$). To keep the same baseline, we did not perform tuning for individual algorithms; all parameters, including minibatch/sample size, is the same and follows Abadi *et al.* [3].

Impact concern: Our work focuses on deploying DP algorithms to build a secure data analysis system while achieving strong privacy guarantees in the view of practical constraints. By designing oblivious sampling algorithms we can, at the same time, make use of the TEEs and the rigorous DP analysis based on sampling.

[Meta-Review · NeurIPS 2019]

The paper offers algorithms for oblivious sampling for application in differentially private data analysis. The algorithms are simple but the analyses are non-trivial and the results are new. The proposed techniques provide a useful and practical solution for implementing DP algorithms involving privacy amplification by sampling. On the other hand, as pointed out by the reviewers, the experimental component is rather limited and can be improved.